# Characterization, Variables, and Antioxidant Activity of the Maillard Reaction in a Fructose–Histidine Model System

**DOI:** 10.3390/molecules24010056

**Published:** 2018-12-24

**Authors:** Pengli Liu, Xiaoming Lu, Ningyang Li, Zhenjia Zheng, Xuguang Qiao

**Affiliations:** Key Laboratory of Food Processing Technology and Quality Control in Shandong Province, College of Food Science and Engineering, Shandong Agricultural University, Tai’an 271018, China; milly.liu@163.com (P.L.); xxalxm@126.com (X.L.); ningyangli@126.com (N.L.)

**Keywords:** Maillard reaction, model system, fructose, characterization, variables, antioxidant activity

## Abstract

Fructose and its polysaccharides are widely found in fruits and vegetables, with the Maillard reaction of fructose affecting food quality. This study aimed to investigate the Maillard reaction of fructose using a fructose–histidine model system. The reaction process was characterized using fluorescence spectroscopy and ultraviolet spectroscopy. The effects of temperature, initial reactant concentration, initial fructose concentration, initial histidine concentration, and initial pH value on the different stages of the Maillard reaction were studied. Reactant reduction, ultraviolet and fluorescence spectra, acetic acid content, 5-hydroxymethylfurfural (5-HMF) content, and browning intensity were evaluated. The results showed that increasing the temperature and reactant concentration promoted the condensation reaction of fructose and amino acid in the early stage, the formation of intermediate products with ultraviolet absorption and fluorescence in the intermediate stage, and the formation of pigment in the final stage. The 5-HMF concentration decreased with increasing histidine concentration and initial pH value. Changes in the shape of ultraviolet and fluorescence spectra showed that the initial pH value affected not only the reaction rate, but also the intermediate product types. The 1,1-diphenyl-2-picrylhydrazyl (DPPH) scavenging rate of the Maillard reaction products increased with increasing temperature, reactant concentration, and initial pH value.

## 1. Introduction

The Maillard reaction is a series of continuous and complex reactions starting with the condensation of carbonyl compounds (usually reducing sugars) and amino compounds (such as proteins, peptides, and amino acids), which generates various products, including 5-hydroxymethylfurfural (5-HMF), organic acids, dicarbonyl compounds, nitrogen-containing heterocycles, and macromolecular pigments (melanoidins) [1,2,3]. The Maillard reaction was discovered by Louis Camille Maillard in a heated solution containing glucose and glysine [4], after which Hodge divided the reaction process into three stages, namely, the early, intermediate, and final stages [5]. The Maillard reaction is influenced by many factors, including temperature [6], pH [7], reactant type [8], reactant concentration [9], buffer salts [10], and metal ions [11].

During food processing and storage, the Maillard reaction has a major impact on the color, flavor, and nutritional value of products, and can form a desirable candy flavor, baked aroma, and golden color, as well as generate antioxidant substances [12,13]. However, the Maillard reaction also has many disadvantageous effects, including causing unwanted color changes and odors, loss of proteins and amino acids, and the production of toxic substances, such as acrylamide [14,15]. Therefore, elucidating and regulating the mechanism of the Maillard reaction is important for improving food quality.

Owing to the complexity of food ingredients, the Maillard reaction is generally studied in a model system consisting of a reducing sugar and amino acid. Most previous research has used glucose as the carbonyl donor. Studies of the fructose Maillard reaction are relatively limited, with no previous reports having systematically researched the variables of this reaction. Fructose and its polysaccharides are widely found in garlic, onion, chicory, and grain coffee [16,17]. As fructans account for 70%–80% of garlic dry matter [18], the quality of its hot processing products, such as black garlic, might be affected by the fructose Maillard reaction [19,20]. Furthermore, the wide use of high-fructose corn syrups means that the Maillard reaction of fructose and amino acids occurs widely in food processing and, therefore, should not be ignored.

In this study, the Maillard reaction of fructose and histidine was systematically studied. Fluorescence spectroscopy and ultraviolet spectroscopy were used to characterize the reaction process. The effects of temperature, initial reactant concentration, initial fructose concentration, initial histidine concentration, and initial pH value on this Maillard reaction were investigated, and the antioxidant activities of the Maillard reaction products were studied. This study provides a foundation for further study of the fructose Maillard reaction mechanism and provides a theoretical basis for quality control in food processing.

## 2. Results and Discussion

### 2.1. Characterization of Fructose–Histidine Maillard Reaction Model System

#### 2.1.1. Ultraviolet Spectra

In the intermediate stage of the Maillard reaction, some products that show ultraviolet absorption are generated [21]. Figure 1 shows the ultraviolet spectra of the fructose–histidine Maillard reaction model system during heating at 80 °C and pH 6.0 for up to 15 days. Before heating, the maximum absorption peak appeared at 217 nm, with no absorbance detected in the wavelength range of 240–400 nm. With prolonged heating, the absorbance at 217 nm increased, while an absorbance at 240–400 nm was detected and increased over time. This enhanced ultraviolet absorption was attributed to the formation of new compounds in the Maillard reaction of fructose and histidine.

#### 2.1.2. Fluorescence Spectra

During the Maillard reaction, some fluorescent compounds are produced before pigment formation [22]. Figure 2 shows the fluorescence spectra of the fructose–histidine Maillard reaction model system during heating. Before heating, the model system showed a fluorescence band centered at 348 nm, which was identified as the characteristic reactant peak. After heating, the model system showed a fluorescence band centered at 398 nm, which was attributed as characteristic peak of the fluorescent products of the Maillard reaction of fructose and histidine. As shown in Figure 2, as the heating time increased, the fluorescence intensity at 348 nm decreased and the fluorescence intensity at 398 nm increased. The changes in fluorescence intensity of the two characteristic peaks indicated that the reactants were gradually consumed and fluorescent products were generated during heating. The fluorescence intensity at 398 nm reached a maximum value after 6 days and then decreased with prolonged heating time up to 15 days due to conversion of the fluorescent products to pigment. A similar observation was reported by Jiang for the glucose-tyramine model system, in which the fluorescence intensity at 420 nm increased significantly and then decreased with prolonged heating [23].

### 2.2. Effect of Variables on Reactant Concentration

It was observed from Figure 3 that fructose and histidine reduction increased with increasing temperature, initial reactant concentration, initial fructose concentration, initial histidine concentration, and initial pH value.

Fructose reduction increased from 4.043 ± 0.679 mg at 50 °C to 99.381 ± 6.243 mg at 90 °C, while histidine reduction rose from 5.200 ± 1.414 mg at 50 °C to 78.067 ± 4.027 mg at 90 °C. The results confirmed that higher temperatures accelerated the rate of the condensation reaction between fructose and histidine in the early stage of the fructose–histidine Maillard reaction. Similar results have been reported previously for the glucose–tyramine Maillard reaction model system, in which the tyramine reduction rate increased as the temperature was increased from 60 °C to 110 °C [23]. Furthermore, based on the data above, the effect of temperature on fructose reduction was greater than that on histidine reduction, meaning that a high temperature can induce other reactions of fructose without histidine being involved.

In the equimolar reactant system, as the reactant concentration was increased from 0.02 M to 0.10 M, fructose reduction increased from 4.594 ± 0.465 mg to 30.299 ± 1.618 mg and histidine reduction increased from 5.434 ± 0.989 mg to 35.000 ± 1.697 mg. In the excess fructose system, as the fructose concentration was increased from 0.02 M to 0.16 M, fructose reduction increased from 3.249 ± 0.531 mg to 23.953 ± 0.503 mg, while histidine reduction increased from 5.867 ± 1.815 mg to 13.200 ± 0.283 mg. There was no significant change in histidine reduction as fructose concentration was increased from 0.04 M to 0.10 M (*p* > 0.05), which meant that the effect of fructose concentration on fructose reduction was greater than that on histidine reduction. In the excess histidine system, as the histidine concentration was increased from 0.02 M to 0.16 M, fructose reduction increased from 3.757 ± 0.706 mg to 11.132 ± 0.783 mg, while histidine reduction increased from 5.302 ± 1.766 mg to 30.900 ± 7.495 mg. Therefore, increasing the reactant concentration clearly promoted the condensation rate of fructose and histidine. Furthermore, excess fructose induced other fructose reactions and excess histidine induced other histidine reactions.

Using initial pH values of 3.0, 4.0 and 5.0, the reactant reductions were small and showed no marked variation (*p* > 0.05). For initial pH values of 6.0, 7.0, 8.0, 9.0, and 10.0, as the initial pH value was increased, both fructose and histidine reduction increased. Fructose reduction increased more significantly than histidine reduction, from 35.139 ± 1.024 mg at pH 6.0 to 156.679 ± 2.091 at pH 10.0. These results were in agreement with those of Ajandouz et al., who reported that heating a mixture of fructose and lysine at 100 °C at different pH values resulted in increased fructose and lysine reduction with increasing pH value [24]. One reason for the increase in reactant reductions may be caramelization of fructose which is accelerated under alkaline condition. In the Maillard reaction with glucose as carbonyl donor, the pH value affects the glucose and amino acid structures. Open-chain glucose and deprotonated amino acids, which are regarded as the active forms in the Maillard reaction, are favored at higher pH values [25,26]. Therefore, a high pH accelerated the condensation of amino groups with carbonyl groups in the early stage of the Maillard reaction and promoted glucose and amino acid reduction. The effect of the initial pH value on the Maillard reaction of fructose might be explained similarly. Amino acid participates in the condensation reaction, forming Amadori- or Heyns-type compounds, in the early stage of Maillard reaction, while in the intermediate stage, degradation of the Amadori-type compounds leads to amino acid regeneration [27,28]. This resulted in the histidine reduction being smaller than the fructose reduction.

### 2.3. Effect of Variables on Ultraviolet Spectra and Absorbance at 294 nm

As mentioned previously, products with ultraviolet absorption are formed in the intermediate stage of the Maillard reaction, with the absorbance at 294 nm usually used to indicate these colorless compounds [9]. As shown in Figure 4A–D, the absorbance at 294 nm was enhanced with increasing temperature, reactant concentration, fructose concentration, and histidine concentration. The results also indicated that increasing the temperature, fructose concentration, and histidine concentration promoted the degradation of the Heyns-type compounds formed in the early stage and the generation of intermediate products with ultraviolet absorption.

As shown in Figure 4E, the change in initial pH value affected not only the ultraviolet absorbance, but also the ultraviolet spectrum shape. When the initial pH value was ≤5.0, three absorption peaks were observed at 217, 283, and 340 nm, with the absorbances at 283 and 340 nm decreasing with increasing initial pH value. When the initial pH value was 6.0 or 7.0, the peaks at 283 nm and 340 nm disappeared. When the initial pH value was ≥8.0, the shape of the ultraviolet spectrum changed, with a new peak observed at 266 nm. The difference in spectrum shape was caused by different products, which suggested that the Heyns-type compounds was degraded though different path in acid and alkaline conditions. Changes in the ultraviolet spectra indicated that increasing the initial pH value promoted the formation of intermediate products with ultraviolet absorbance and changed the species produced.

### 2.4. Effect of Variables on Fluorescence Spectra

As shown in Figure 5, all fluorescence spectra showed similar shapes, except at pH 7.0, 8.0, 9.0, and 10.0.

Figure 5A shows that the fluorescence intensity at 398 nm increased with increasing temperature from 50 °C to 70 °C. This indicated that increasing the temperature promoted the formation of fluorescence compounds in the intermediate stage of the Maillard reaction of fructose and histidine. As the temperature continued rising to 90 °C, the fluorescence intensity at 398 nm decreased. The weakened fluorescence intensity was due to the fluorescence compounds being transformed into melanoidin in the final stage of the Maillard reaction. Therefore, high temperature had a greater effect on the reaction rate of the final stage than on the middle stage of the Maillard reaction between fructose and histidine. Jiang examined the impact of temperature on the fluorescence intensity at an emission wavelength of 425 nm in galactose–bovine casein peptide Maillard reaction products, finding that the fluorescence intensity reached a maximum value at 100 °C and gradually decreased at higher temperatures [7].

As shown in Figure 5B–D, increasing the fructose concentration or histidine concentration promoted the formation of fluorescence compounds.

Figure 5E shows that the initial pH value affected not only the fluorescence intensity, but also the shape of the fluorescence spectrum. When the initial pH value was ≤6.0, the strongest peak of the fructose–histidine Maillard reaction model system occurred at an emission wavelength of 398 nm, while the fluorescence intensity at 398 nm initially increased and then decreased with increasing initial pH value. When the initial pH value was ≥7.0, the fluorescence intensity at 398 nm decreased as the pH increased, with the strongest peak moving to 470 nm. The fluorescence intensity at 470 nm was not much affected by changing pH. The change in the shape of the fluorescence spectrum indicated that the fluorescence compound type had changed.

### 2.5. Effect of Variables on pH and Acetic Acid Content

Owing to the formation of organic acids in the intermediate stage of the Maillard reaction, the pH value drops during this stage [29]. Figure 6 shows the effect of temperature, initial reactant concentration, initial fructose concentration, initial histidine concentration, and initial pH value on this pH drop and the acetic acid content.

As the temperature was increased, the pH drop and acetic acid content became larger, indicating that a higher temperature promoted acetic acid formation in the intermediate stage of the Maillard reaction of fructose and histidine. These results were in accordance with those of Lan et al., who found that the pH value of the xylose–soybean peptide system decreased with increasing temperature [30].

The pH drop showed no significant change (*p* > 0.05) as the initial reactant concentration was increased from 0.02 M to 0.10 M, but the acetic acid content increased greatly from 0.624 ± 0.070 mg/L to 7.730 ± 0.684 mg/L. These results indicated that increasing the initial reactant concentration promoted acetic acid formation in the intermediate stage of the fructose–histidine Maillard reaction, and that the pH stability was provided by histidine buffering. The acetic acid content increased with increasing fructose concentration and increasing histidine concentration in their respective excess systems. However, the pH drop in the excess histidine system stayed near zero and showed no significant change (*p* > 0.05) regardless of the histidine concentration, again owing to buffering by histidine.

Increasing the initial pH value also increased the pH drop and acetic acid content of the fructose–histidine Maillard reaction model system, indicating that a high pH value promoted formation of acetic acid formation in the intermediate stage of the Maillard reaction of fructose and histidine. An increase in acetic acid content with increasing pH was also reported for the system containing xylose and glycine heated at 120 °C [31]. Notably, when the initial pH value was increased from 9.0 to 10.0, the pH drop increased, but the acetic acid content decreased. This might be due to other organic acids formed under the strong alkaline conditions.

### 2.6. Effect of Variables on 5-HMF Content

5-HMF is generated in the intermediate stage of the Maillard reaction and consumed in the final stage through participation in pigment formation. As shown in Figure 7, the 5-HMF content increased with increasing temperature, initial reactant concentration, and initial fructose concentration, but decreased with increasing initial histidine concentration and initial pH value.

No 5-HMF was detected in the fructose–histidine Maillard reaction model system heated at 50 °C and 60 °C for 5 days. As the temperature was increased further, the 5-HMF content increased from 0.816 ± 0.033 μg/mL at 70 °C to 18.066 ± 1.513 μg/mL at 90 °C. This indicated that high temperatures promoted 5-HMF formation in the intermediate stage of the Maillard reaction of fructose and histidine. A previous study also found that the 5-HMF content was promoted at higher temperatures in the glucose-wheat flour system [32].

In the equimolar reactant system, the 5-HMF content increased from 0.705 ± 0.132 μg/mL to 3.391 ± 0.383 μg/mL as the reactant concentration increased from 0.02 M to 0.10 M. In the excess fructose system, increasing the fructose concentration from 0.02 M to 0.16 M caused the 5-HMF content to increase sharply from 0.958 ± 0.185 μg/mL to 26.332 ± 1.618 μg/mL, indicating that fructose promoted 5-HMF formation in the intermediate stage of the Maillard reaction of fructose and histidine. This was in agreement with the results of Chériot et al., who heated 0.25 M cysteine with glucose of various concentrations (0–1.00 M) and found that a higher glucose resulted in more 5-HMF [33]. In the excess histidine system, increasing the histidine concentration from 0.02 M to 0.16 M decreased the 5-HMF content from 0.796 ± 0.060 μg/mL to 0.089 ± 0.018 μg/mL. This decrease in 5-HMF content was attributed to the pigment formation involving 5-HMF and histidine in the final stage of the Maillard reaction of fructose and histidine, while an increased histidine content increased the reaction rate.

As the initial pH value was increased from 3.0 to 9.0, the 5-HMF content decreased from 329.938 ± 6.713 μg/mL to 1.596 ± 0.224 μg/mL, indicating that a high pH promoted 5-HMF consumption in the final stage of the Maillard reaction of fructose and histidine.

### 2.7. Effect of Variables on Browning Intensity

The absorbance at 420 nm (A420) is often used to indicate the browning intensity caused by pigment formation in the final stage of the Maillard reaction [7]. As shown in Figure 8, the browning intensity of the fructose–histidine Maillard reaction model system increased with increasing temperature, initial reactant concentration, initial fructose concentration, initial histidine concentration, and initial pH value.

A dramatic increase in A420 in the fructose–histidine Maillard reaction model system was observed as the temperature was increased from 50 °C to 90 °C. This indicated that high temperatures accelerated pigment formation in the final stage of the Maillard reaction of fructose and histidine. These results were in agreement with the work of Vhangani et al., who reported a sharp increase in A420 with increasing temperature in ribose–lysine and fructose–lysine systems [6].

The A420 also increased with increasing reactant concentration. In the excess fructose system, the A420 increased from 0.190 ± 0.025 to 0.755 ± 0.023 as the fructose concentration increased from 0.02 M to 0.16 M, while in the excess histidine system, the A420 increased from 0.248 ± 0.016 to 0.623 ± 0.053 as the histidine concentration increased from 0.02 M to 0.16 M. These data indicated that fructose more markedly promotes pigment generation compared with histidine.

The A420 increased 3.86-fold as the initial pH value was increased from 3.0 to 10.0, which showed that a high pH value was beneficial for pigment formation in the final stage of the Maillard reaction of fructose and histidine. Similar results have been reported for the glucose–ammonium sulfate system, which showed a significant increase in A420 with increasing pH [34].

### 2.8. Effect of Variables on Antioxidant Activity

The 1,1-diphenyl-2-picrylhydrazyl (DPPH) scavenging assay is widely used to determine the antioxidant activity of Maillard reaction products [35]. Figure 9 shows the effect of temperature, initial reactant concentration, initial fructose concentration, initial histidine concentration, and initial pH value on the DPPH scavenging rate of the Maillard reaction products from the fructose–histidine system. As the temperature was increased from 50 °C to 90 °C, the DPPH scavenging rate increased significantly from 8.86 ± 1.91% to 92.42 ± 2.13%. Increasing the reactant concentration also increased the DPPH scavenging rate of the Maillard reaction products, with the fructose concentration having a more significant impact than the histidine concentration. The DPPH scavenging rate also increased with increasing initial pH value. The Maillard reaction products generated under alkaline conditions showed stronger antioxidant activity than those generated under acidic conditions.

## 3. Materials and Methods

### 3.1. Chemicals

l-Histidine, fructose, and hydrochloric acid (36%–38%) were purchased from Sinopharm Chemical Technology Co., Ltd. (Shanghai, China). Disodium phosphate and sodium hydroxide were purchased from Kaitong Chemical Technology Co., Ltd. (Tianjin, China). Ethanol was purchased from Yongda Chemical Technology Co., Ltd. (Tianjin, China). Acetonitrile and methanol of HPLC grade were purchased from Yuwang Chemical Technology Co., Ltd. (Shandong, China). Acetic acid standard, 5-HMF standard, and DPPH were purchased from Sigma (St Louis, MO, USA). All other chemicals used in this study were of analytical grade.

### 3.2. Preparation of Fructose–Histidine Maillard Reaction Model System

Unless otherwise stated, the model system contained 0.10 M fructose and 0.10 M histidine in a total volume of 10.0 mL at pH 6.0, and was heated at 80 °C in an oven for 5 days. After heating, the model system was cooled to room temperature to terminate the reaction.

### 3.3. Variables of the Fructose–Histidine Maillard Reaction Model System

#### 3.3.1. Temperature

To evaluate the effect of temperature on the Maillard reduction of fructose and histidine, the model reaction system was heated at 50, 60, 70, 80, and 90 °C for 5 days.

#### 3.3.2. Initial Reactant Concentration

To evaluate the effect of initial reactant concentration on the Maillard reaction of fructose and histidine, the model reaction system containing equimolar fructose and histidine concentrations of 0.02, 0.04, 0.06, 0.08, and 0.10 M were incubated at 80 °C and pH 6.0 for 5 days.

#### 3.3.3. Excess Fructose

To evaluate the effect of fructose concentration on the Maillard reaction of fructose and histidine, the model reaction system containing 0.02 M histidine and 0.02, 0.04, 0.08, 0.12, and 0.16 M fructose was incubated at 80 °C and pH 6.0 for 5 days.

#### 3.3.4. Excess Histidine

To evaluate the effect of histidine concentration on the Maillard reaction of fructose and histidine, the model reaction system containing 0.02 M fructose and 0.02, 0.04, 0.08, 0.12, and 0.16 M histidine was incubated at 80 °C and pH 6.0 for 5 days.

#### 3.3.5. Initial pH Value

To evaluate the effect of initial pH value on the Maillard reaction of fructose and histidine, the model reaction system at pH 3.0, 4.0, 5.0, 6.0, 7.0, 8.0, 9.0, and 10.0 was heated at 80 °C for 5 days.

### 3.4. Fructose Concentration Determination

Fructose concentration was determined by HPLC [36], using a Shimadzu RID-10A (Shimadzu Co., Tokyo, Japan) refractive index detector and an Agilent ZORBAX carbohydrate analysis column (5 μm, 4.6 × 250 mm, Agilent, Santa Clara, CA, USA). The mobile phase consisted of acetonitrile/water (80:20, *v*/*v*) at a flow rate of 1.0 mL/min. The column temperature was 40 °C and a 15-μL sample was injected into the HPLC system. Data analysis was performed using LC Solution software. Calibration curves were constructed for peak area vs. standard fructose concentration.

### 3.5. Histidine Concentration Determination

The histidine concentration was determined using an automatic Amino Acid Analyzer (L-8900, Hitachi, Ltd., Tokyo, Japan).

### 3.6. Ultraviolet Spectra

Ultraviolet spectra were recorded using a UV–visible spectrophotometer (UV-2450, Shimadzu Co., Tokyo, Japan). Samples were diluted 100-fold with ultrapure water for measurement. Ultraviolet spectra were recorded at 200–400 nm. The absorbance at 294 nm was also measured.

### 3.7. Fluorescence Spectra

Fluorescence intensities were determined using a fluorescence spectrophotometer (LUMINA, Thermo Fisher Scientific, Waltham, MA, USA). Samples were diluted 20-fold with ultrapure water for measurement. The fluorescence intensities were measured at an excitation wavelength of 347 nm and emission wavelength of 300–600 nm.

### 3.8. Determination of pH Value

The pH value of the model system was monitored using an Orion 868 pH meter (Thermo Fisher Scientific, Waltham, MA, USA).

### 3.9. Acetic Acid Concentration Determination

The acetic acid concentration was determined by HPLC [15] using a Shimadzu SPD-M20A (Shimadzu Co., Tokyo, Japan) diode array detector and an Agilent ZORBAX SB-C18 column (5 μm, 9.4 × 250 mm). Disodium phosphate buffer (10 mM) acidified to pH 2.7 with phosphoric acid and acetonitrile (90:10, *v*/*v*) was used as the mobile phase at a flow rate of 0.8 mL/min. The column temperature was 30 °C and a 10-μL sample was injected into the HPLC system. The UV detection wavelength was set at 210 nm. Data analysis was performed using LC Solution software. Calibration curves were constructed for peak area vs. standard acetic acid concentration.

### 3.10. 5-HMF concentration determination

5-HMF determination was performed by HPLC [37] using a Shimadzu SPD-M20A diode array detector and an Agilent ZORBAX SB-C18 column (5 μm, 9.4 × 250 mm). The mobile phase consisted of water/methanol (90:10, *v*/*v*) at a flow rate of 1.0 mL/min. The column temperature was 40 °C and a 10-μL sample was injected into the HPLC system. The UV detection wavelength was set at 285 nm. Data analysis was performed using LC Solution software.

### 3.11. Measurement of Browning Intensity

The browning intensity of the model system was determined by measuring the absorbance at 420 nm against water using a UV–Vis spectrophotometer (Persee UV T6, Persee Co., Beijing, China) [8]. The solution was diluted when necessary to obtain optical absorbance.

### 3.12. DPPH Scavenging Activity Determination

The DPPH scavenging activity was determined using the method of Yamabe et al. [38]. The sample solution (2 mL) was mixed with DPPH solution (2 mL, 0.1 mmol/L) and placed in the dark at 37 °C for 1 h. The absorbance at 517 nm (denoted as A) was then measured using a UV-Visible spectrophotometer (Persee UV T6, Persee Co., Beijing, China). A mixture of distilled water and DPPH was measured as the control (denoted as A_s_), while a mixture of sample solution and ethyl alcohol was measured as the blank (denoted as A_0_). The DPPH scavenging rate was calculated using the formula shown in Equation (1):(1)$$\mathrm{DPPH}\;\mathrm{scavenging}\;\mathrm{rate}=\left(1-\frac{A-A_{0}}{A_{s}}\right)\times 100\%$$

### 3.13. Statistical Analyses

Statistical analyses were performed using IBM SPSS Statistics 22.0 software. Data were analyzed by one-way analysis of variance (ANOVA) followed by Duncan’s multiple range test. P values of <0.05 were considered statistically significant. All experiments were conducted in triplicate and data were expressed as means ± standard deviation.

## 4. Conclusions

In the Maillard reaction of fructose and histidine, intermediate products with ultraviolet absorption were generated, as characterized by ultraviolet spectroscopy. Fluorescence spectra showed that fluorescent products were also generated and later converted into pigment. In the early stage of the Maillard reaction, the condensation of fructose and histidine was promoted by increasing the temperature, reactant concentration, and initial pH value. In the intermediate stage of the Maillard reaction, the formation of products with ultraviolet absorption and fluorescent properties, as well as acetic acid, was promoted by increasing the temperature, reactant concentration, and initial pH value. The initial pH value changed the species of the intermediate products. 5-HMF formation was accelerated by increasing the temperature and fructose concentration. In the final stage of the Maillard reaction, the browning intensity was enhanced by increasing the temperature, reactant concentration, and initial pH value. Increasing the histidine concentration and initial pH value facilitated 5-HMF consumption, while increasing the temperature promoted the conversion of fluorescent intermediate products into pigment. The fructose–histidine Maillard reaction products showed DPPH scavenging activity, which was enhanced by increasing the temperature, reactant concentration, and initial pH value.

## Figures and Tables

**Figure 1 molecules-24-00056-f001:**
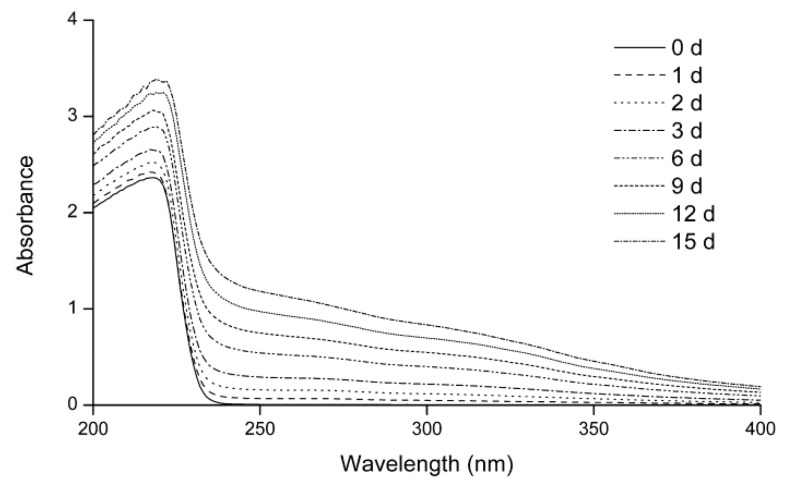
Ultraviolet spectra of the fructose–histidine Maillard reaction model system at pH 6.0 and 80 °C for up to 15 days.

**Figure 2 molecules-24-00056-f002:**
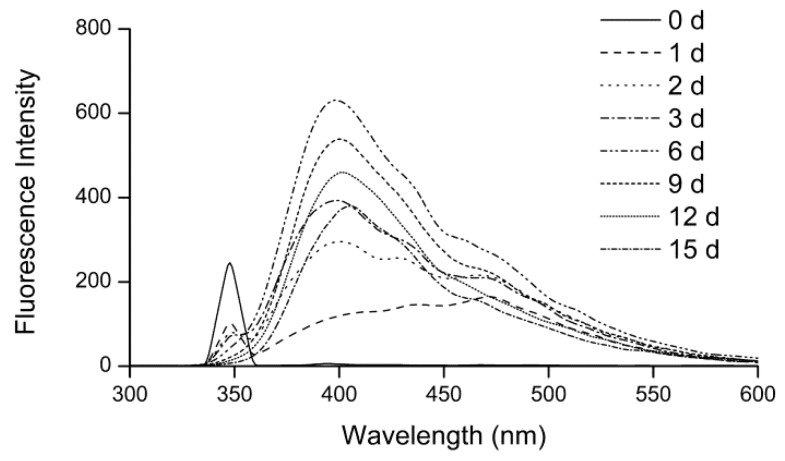
Fluorescence spectra of the fructose–histidine Maillard reaction model system at pH 6.0 and 80 °C for up to 15 days.

**Figure 3 molecules-24-00056-f003:**
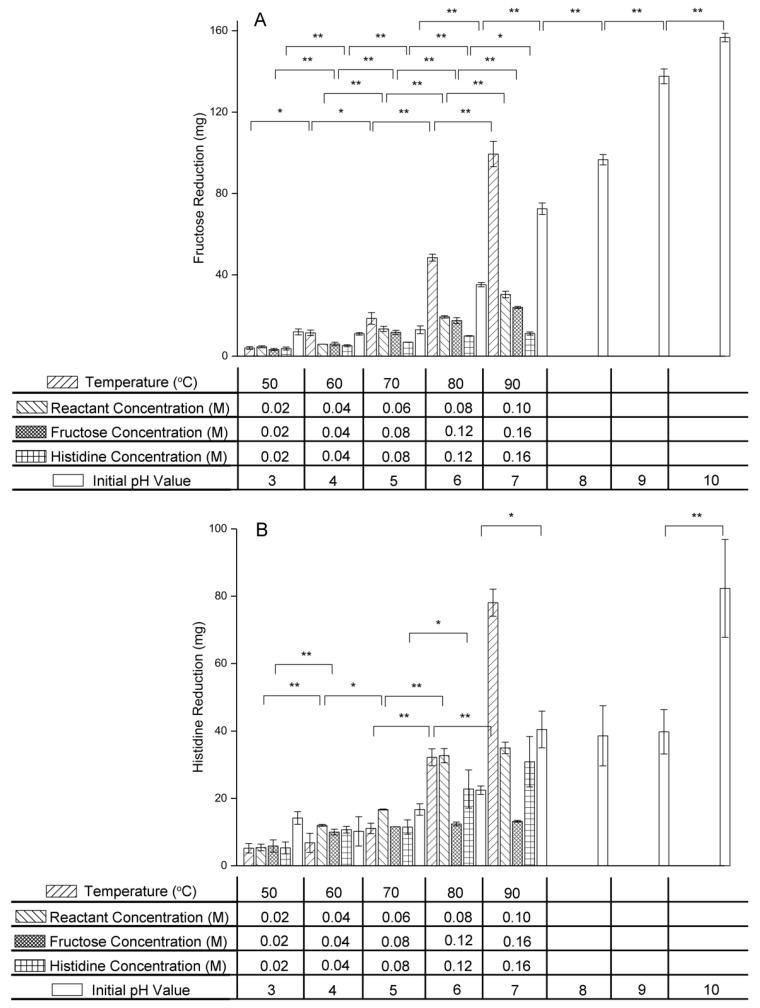
(**A**) Fructose and (**B**) histidine reduction in the fructose–histidine Maillard reaction model system (* *p* < 0.05; ** *p* < 0.01).

**Figure 4 molecules-24-00056-f004:**
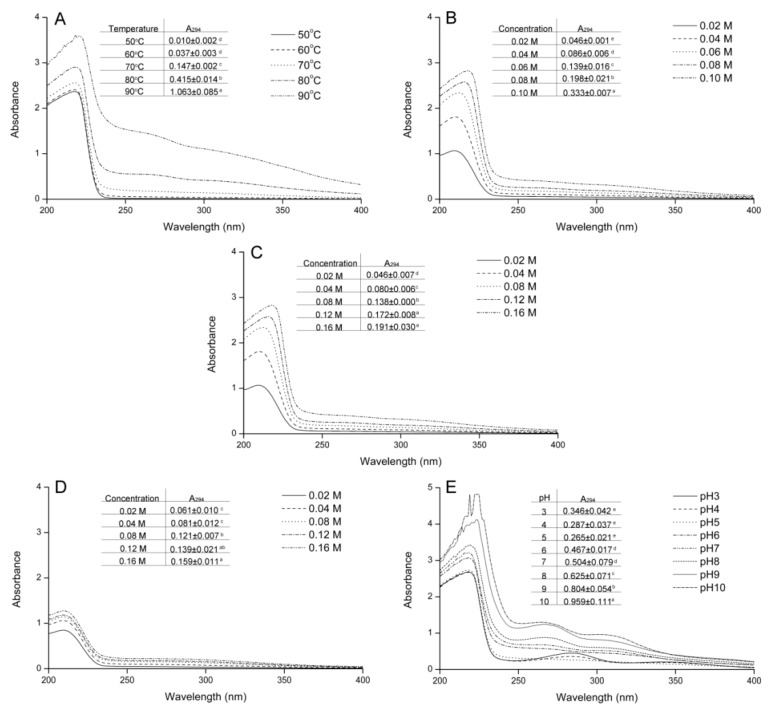
Effect of (**A**) temperature, (**B**) initial reactant concentration, (**C**) initial fructose concentration, (**D**) initial histidine concentration, and (**E**) initial pH value on the ultraviolet spectrum and absorbance at 294 nm (insert table) of the fructose–histidine Maillard reaction model system heated for 5 days. Data in the same table with different letters are significantly different (*p* < 0.05).

**Figure 5 molecules-24-00056-f005:**
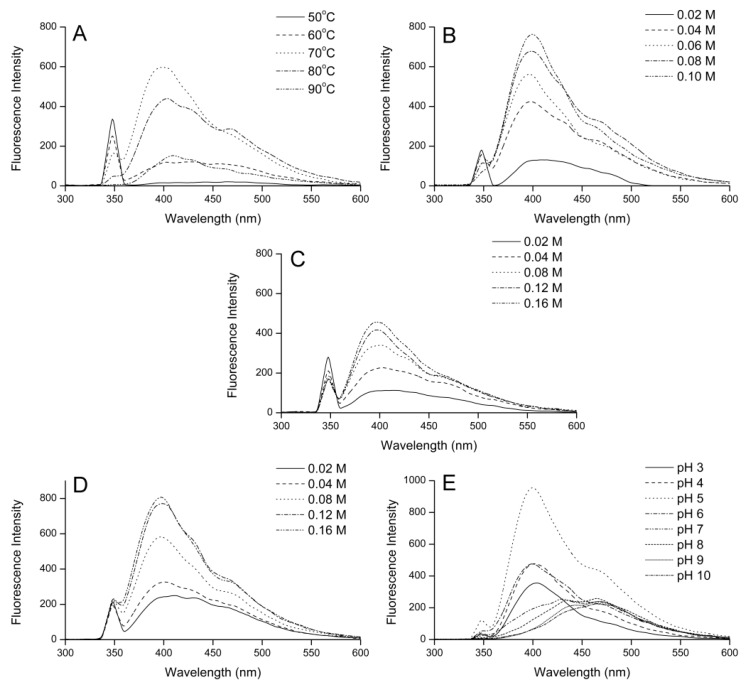
Effect of (**A**) temperature, (**B**) initial reactant concentration, (**C**) initial fructose concentration, (**D**) initial histidine concentration, and (**E**) initial pH value on the fluorescence spectrum of the fructose–histidine Maillard reaction model system heated for 5 days.

**Figure 6 molecules-24-00056-f006:**
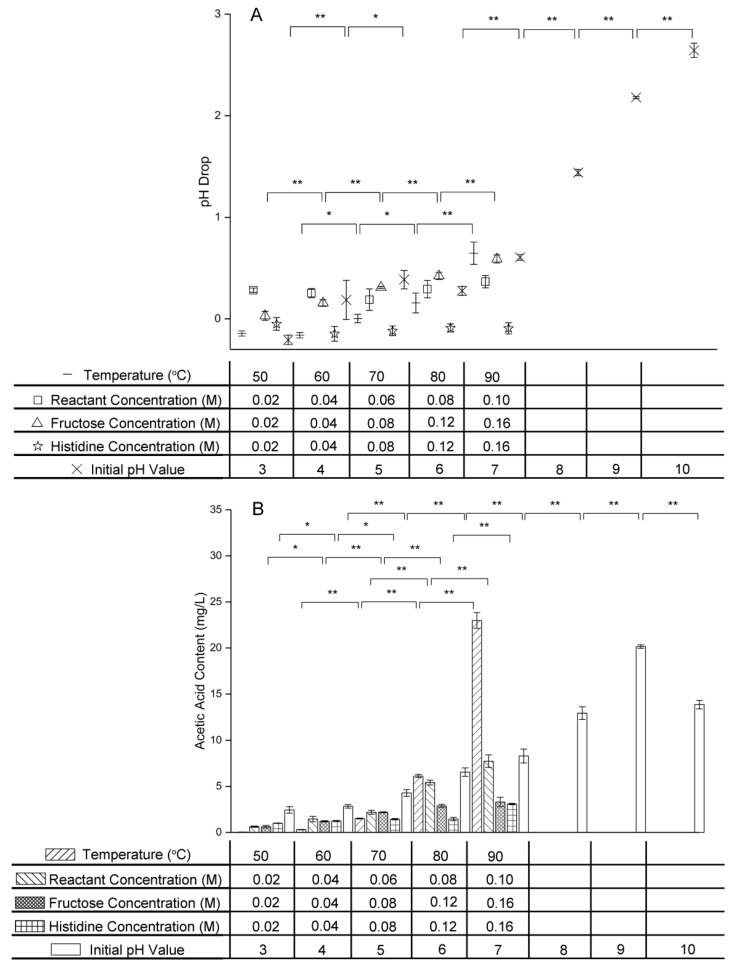
Changes in (**A**) pH drop and (**B**) acetic acid content of the fructose–histidine Maillard reaction model system heated for 5 days (* *p* < 0.05; ** *p* < 0.01).

**Figure 7 molecules-24-00056-f007:**
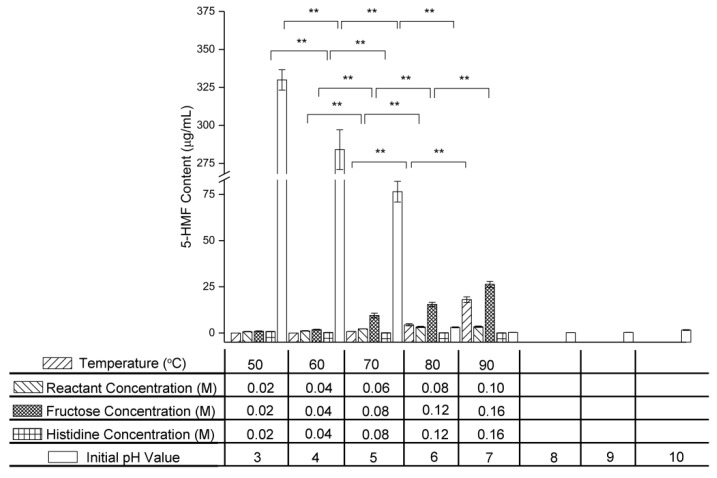
Changes in 5-HMF content of the fructose–histidine Maillard reaction model system heated for 5 days (** *p* < 0.01).

**Figure 8 molecules-24-00056-f008:**
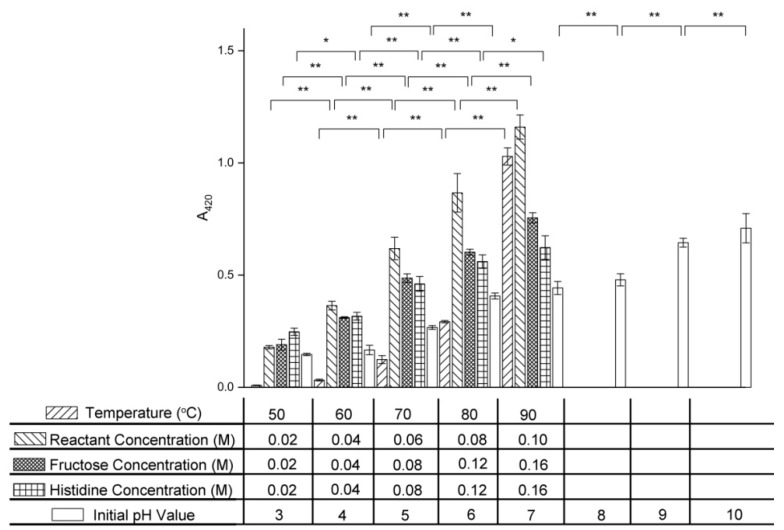
Changes in absorbance at 420 nm of the fructose–histidine Maillard reaction model system heated for 5 days (* *p* < 0.05; ** *p* < 0.01).

**Figure 9 molecules-24-00056-f009:**
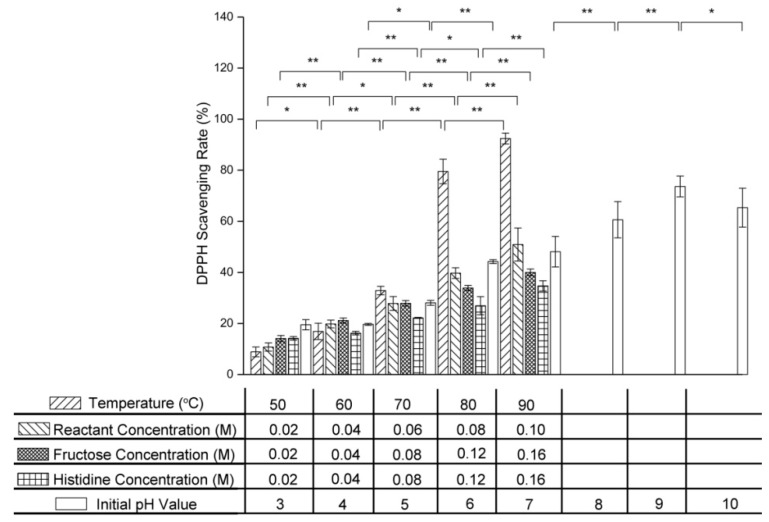
Changes in DPPH scavenging ability of products from the fructose–histidine Maillard reaction model system heated for 5 days (* *p* < 0.05; ** *p* < 0.01).

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
