# Peer review of "Characterization, Variables, and Antioxidant Activity of the Maillard Reaction in a Fructose–Histidine Model System"

_molecules, 2018, doi:10.3390/molecules24010056_

Reviewer 1 Report

The authors studied the investigate the Maillard reaction of fructose using a fructose-histidine model system. They reported some new findings. Overall, this is a well-written and well-organized research article.

The data in Results should be interpreted more statistically. If there were significant differences among treatment, P<0.05 should be added. If not, P>0.05 should be added.

Line 25-26: The results should be described in Abstract.

Line 293-295: Any references for developing this fructose-histidine Maillard reaction model system?

Line 353-354: References should be cited for measuring browning intensity.

Author Response

Response to Reviewer 1 Comments

Point 1: The data in Results should be interpreted more statistically. If there were significant differences among treatment, P<0.05 should="" be="" added.="" if="" p="">0.05 should be added.

Response 1: The date in Results has been interpreted more statistically. Significant P values have been added and indicated in figures.

Point 2: Line 25-26: The results should be described in Abstract.

Response 2: The results have been described in Abstract.

Point 3: Line 293-295: Any references for developing this fructose-histidine Maillard reaction model system?

Response 3: This fructose-histidine Maillard reaction was developed according to our previous study. We heated garlic neutral polysaccharide hydrolysate (the major ingredient is fructose) with 17 kinds of amino acids and found the model solution containing histidine had the greatest browning intensity.

Point 4: Line 353-354: References should be cited for measuring browning intensity.

Response 4: Reference has been cited for measuring browning intensity.

Reviewer 2 Report

The manuscript describes the Maillard reaction model and concerns the condensation of fructose and histidine. The problem has been presented in a way that does not bring new information. The  variables used in the study were repeated many times in other similar model studies. The obtained trends of changes resulting from the increase of temperature or pH are well known for the Maillard reaction. The results have been described, but their scientific discussion is lacking. The most important thing is therefore to add the proper interpretation of the results and to determine by more specific analytical techniques the chemical structures that arise in the subsequent stages of the Maillard reaction between the tested carbohydrate and the amino acid. Other comments are below, but their improvement will only slightly increase the scientific value of the manuscript:

Page 2, line 51: Fructans are also the ingredients of onion, and above all chicory, which is a component of grain coffee.

Tables: Change "reatant" to "reactant".

Page 4, line 128-129: This is a dependence known for decades, caramel is produced by the method with ammonia at elevated pH.

Author Response

Response to Reviewer 2 Comments

Point 1: The results have been described, but their scientific discussion is lacking. The most important thing is therefore to add the proper interpretation of the results and to determine by more specific analytical techniques the chemical structures that arise in the subsequent stages of the Maillard reaction between the tested carbohydrate and the amino acid.

Response 1: Some interpretation of the results was added. We are working on the structural analysis of the products formed in the fructose-histidine Maillard reaction. But due to the variety and complexity of Maillard Reaction Products, there are some difficulties in the isolation and purification process. We will keep working on it. Once the structure of the pigment is clear, the mechanisms of the effect of variables on the Maillard reaction will be interpreted more detailed.

Point 2: Page 2, line 51: Fructans are also the ingredients of onion, and above all chicory, which is a component of grain coffee.

Response 2: We have revised this sentence as “Fructose and its polysaccharides are widely found in garlic, onion, chicory, and grain coffee”, and two articles were cited.

Point 3: Tables: Change "reatant" to "reactant".

Response 3: "reatant" has been changed to "reactant" in Tables (Figures 3, 6, 7, 8, and 9).

Point 4: Page 4, line 128-129: This is a dependence known for decades, caramel is produced by the method with ammonia at elevated pH.

Response 4: The sentence “One reason for the increase in reactant reductions may be caramelization of fructose which is accelerated under alkaline condition” has been added.

Reviewer 3 Report

This study is welldone. I have only some minor comments on it.  

Figures 3 and 6 are quite complicated. It would be better to show fructose and histidine reductions (Fig. 3) in separate figures. In addition, pleas show pH and acetic acid (Fig.6) data in separete figures.
Figure legends in Figures 3, 6, 7, 8, and 9 are difficult to understand. Units (C, M) shoud be included in the table.
In addition, significant p-values should be indicated in figures.
line 332: Thermo Fisher Scientific, Waltham, MA, USA.

Author Response

Response to Reviewer 3 Comments

Point 1: Figures 3 and 6 are quite complicated. It would be better to show fructose and histidine reductions (Fig. 3) in separate figures. In addition, pleas show pH and acetic acid (Fig.6) data in separete figures.

Response 1: Fructose and histidine reductions (Fig. 3) have been shown in separate figures (Figure 3A and B). pH and acetic acid (Fig. 6) have been shown in separate figures (Figure 6A and B).

Point 2: Figure legends in Figures 3, 6, 7, 8, and 9 are difficult to understand. Units (C, M) shoud be included in the table.

Response 2: Figure legends in Figures 3, 6, 7, 8, and 9 have been revised. Units have been included in the table.

Point 3: In addition, significant p-values should be indicated in figures.

Response 3: Significant p-values have been indicated in Figures 3, 4, 6, 7, 8, and 9.

Point 4: line 332: Thermo Fisher Scientific, Waltham, MA, USA.

Response 4: The sentence has been revised.

Round  2

Reviewer 2 Report

My general remark regarding the low scientific level of work remains valid after the revision of the manuscript by the authors. The editors of the journal perceive the presented research as worth publishing. I can join this opinion, although I still think that the studies described are rather cursory. Other minor errors that I pointed out have been corrected. For this reason, the manuscript can be published in its current form.